# The Different Roles of Structural and Cognitive Social Capital on Oral Health-Related Quality of Life among Adolescents

**DOI:** 10.3390/ijerph20085603

**Published:** 2023-04-21

**Authors:** Jessica Klöckner Knorst, Mario Vianna Vettore, Bruna Brondani, Bruno Emmanuelli, Thiago Machado Ardenghi

**Affiliations:** 1Department of Stomatology, School of Dentistry, Universidade Federal de Santa Maria, Santa Maria 97105-900, Brazil; 2Department of Health and Nursing Sciences, University of Agder, Postbox 422, N-4604 Kristiansand, Norway; 3Department of Pediatric Dentistry and Orthodontics, School of Dentistry, Universidade de São Paulo, São Paulo 05508-000, Brazil

**Keywords:** adolescent, quality of life, oral health, social capital, psychosocial factors

## Abstract

This study evaluated the relationship of structural and cognitive dimensions of social capital with oral health-related quality of life (OHRQoL) among adolescents. This was a cross-sectional study nested in a cohort of adolescents from southern Brazil. OHRQoL was evaluated using the short version of the Child Perceptions Questionnaire 11-14 (CPQ11-14). Structural social capital was measured by attendance of religious meetings and social networks from friends and neighbours. Cognitive social capital was evaluated through trust in friends and neighbours, perception of relationships in the neighbourhood, and social support during hard times. Multilevel Poisson regression analysis was performed to estimate the association between social capital dimensions and overall CPQ11-14 scores; higher scores corresponded to worse OHRQoL. The sample comprised 429 adolescents with a mean age of 12 years. Adolescents who attended religious meetings less than once a month or never presented higher overall CPQ11-14 scores. Adolescents who did not trust their friends and neighbours, those who believe that their neighbours did not have good relationships, and those reporting no support during hard times also presented higher overall CPQ11-14 scores. OHRQoL was poorer in individuals who presented lower structural and cognitive social capital, with the greatest impact related to the cognitive dimension.

## 1. Introduction

Oral health has been considered a multifaceted concept that is continuously influenced by the values and attitudes of people and communities, reflecting their physical, social, and psychological attributes that are essential to quality of life [1]. Thus, quality of life has been considered an important attribute of patient’s evaluation of their oral health and reorientation of health services [1]. Oral health-related quality of life (OHRQoL) is a multidimensional construct that reflects the extent to which oral disorders affect individuals’ daily functions and well-being [2]. Therefore, oral health emerges as a positive concept highlighting the personal and social resources available to individuals, such as social capital.

Social capital can be defined as the resources embedded in social networks, which can be accessed and mobilized through the existing social ties within or among one’s networks [3,4,5,6]. The previous literature suggests that social capital may be distinguished into structural and cognitive dimensions, which correspond to the quantitative and qualitative aspects of social capital, respectively [3]. The former dimension refers to the extent and intensity of one’s participation in associations or other forms of social activity. The cognitive dimension is less tangible and relates to people’s perception of interpersonal trust, solidarity, and reciprocity. Thus, social capital involves the quantity and quality of accessible resources and benefits among individuals or groups according to their social networks [3,4,5,6].

Social capital and its proxy indicators have been associated with different clinical and subjective oral health measures. The previous literature has shown that individuals with lower social capital were more likely to have dental caries, gingivitis, and worse self-rated oral health and OHRQoL [7,8,9,10]. Previous studies have also verified the relationship of different dimensions of social capital with general health, such as frailty [11], mental health [12], and self-perceived health [13]. Other research has also highlighted the impact of different dimensions of social capital on oral health outcomes in adults and elderly people, including edentulism [14], dental pain [15], and self-perception of oral health [16].

Although previous studies have established the relationship between social capital and oral health outcomes, the possible impact of social capital on OHRQoL during adolescence has been little explored. Adolescence is characterized by biopsychosocial changes, when individuals tend to initiate and establish their own social relationships according to their involvement and engagement in different social groups [17,18]. Further investigations aiming to enhance the understanding of which aspects of social capital are more relevant to oral health in this age group can be useful to develop public health actions to promote oral health and quality of life. Furthermore, elucidating the impact of different aspects of social capital on health has been encouraged by the previous literature [19]. Thus, this study aimed to evaluate the relationship of structural and cognitive social capital with OHRQoL among adolescents. It was hypothesized that adolescents with lower structural and lower cognitive social capital are more likely to report a greater impact on OHRQoL.

## 2. Methods

### 2.1. Study Design and Sample

This was a cross-sectional study nested in a 10-year cohort study with preschool children from the city of Santa Maria, Brazil. Baseline data collection was conducted in 2010 and follow-ups were in 2012, 2017, and 2020. Further methodological details of this cohort study are described elsewhere [10,20]. This study used data from the data collection carried out in 2020.

The baseline sample was obtained during National Children’s Vaccination Day in health centres that were responsible for the vaccination coverage of 90% of children in the city. All healthcare centres with dental offices (15 out of 28 in total) were initially selected as sampling units. Subsequently, every fifth child in the vaccination line was systematically selected and invited to participate. The selected health centres were located in different regions of the city involving populations with different socioeconomic backgrounds. A total of 639 children aged from 1 to 5 years were examined in 2010. Of them, 429 adolescents (response rate = 67.1%) were assessed in 2020.

All individuals who participated in the baseline were invited to participate in 10-year follow-up, with ages ranging from 11 to 15 years old during this period. Data collection of this wave was performed between November 2019 and January 2021, though it was interrupted between March and October 2020 due to the COVID-19 pandemic. Adolescents were evaluated in their schools or households, according to the information available in the database. Participants and their families were contacted via social media (WhatsApp, Instagram, or Facebook) if they were not found using the initial strategy.

The power of the study was calculated in OpenEpi [21] through a post hoc power calculation and considered an alpha error probability of 0.05 and a 95% confidence interval. Overall, adolescents’ CPQ11-14 scores were compared between the non-exposed group (high social capital) and the exposed group (low social capital) according to all social capital variables used, resulting in a study power that ranged from 80 to 100%. 

### 2.2. Data Collection and Variables

All data collection procedures were based on the international standardized criteria for oral health surveys [22,23,24,25]. Data were collected through interviews using structured questionnaires and through oral clinical examinations performed by a trained research team.

OHRQoL was assessed using the Brazilian short-version of Child Perceptions Questionnaire 11-14 (CPQ11-14 ISF:16) [22]. This questionnaire is composed of 16 items, grouped into 4 different domains: oral symptoms, functional limitation, emotional well-being, and social well-being. All questions are responded to considering a Likert scale from 0 to 4 points. The overall CPQ11-14 score is made by the sum of the scores of the 16 items. The overall CPQ11-14 scores were used in the data analysis and can range from 0 to 64 points, with higher scores indicating a lower level of quality of life.

Individual social capital was assessed considering the structural and cognitive dimensions [3]. The structural social capital reflects the extent and intensity of participation in associations or other forms of social activity, whereas the cognitive side reflects people’s perception of interpersonal trust, solidarity, and reciprocity [3,6]. The structural dimension was evaluated according to the social networks with the following questions: “How often do you attend group religious activities?”, and “In the last 12 months, how often have you visited or received visits from friends and neighbours?”, with the possible response options including the following: (0) at least once a month; and (1) less than once a month or never. The cognitive dimension was assessed by the following questions: “Do you think your friends and neighbours can be trusted?”, (0) yes or (1) no; “Do people in your neighbourhood have good relations with each other?”, (0) yes or (1) no; and “If something bad happens to you, would someone help you in this situation?”, (0) yes or (1) no. These items have been commonly used as proxy measures of structural and cognitive social capital according to the previous literature [7,9,10,23,24].

Demographic, socioeconomic, and clinical data were collected as possible confounding factors on the relationship between social capital and OHRQoL [2,10]. Demographic characteristics included sex (girls and boys) and age (in years and dichotomized by the sample mean). Socioeconomic status was assessed according to monthly household income based on the sum of all income sources of the family in the previous month in Brazilian Reais (Brazilian currency). For data analysis, household income was categorized into <1 Brazilian minimum wages (BMW) and ≥1 BMW. One BMW was the equivalent to USD 220 in 2020. Dental caries were assessed with calibrated examiners using the International Caries Detection and Assessment System (ICDAS) criteria [25], in which all dental surfaces were evaluated. Dental caries with cavitated lesions were grouped as present (ICDAS scores 3, 5, and 6) or absent (ICDAS scores 0, 2, and 4). A total of seven examiners conducted clinical examinations. All clinical examiners were dentists who were previously trained and calibrated for ICDAS evaluation, totaling a process of 36 h, including theoretical training, photographic image evaluation, exercise with exfoliated teeth, and clinical evaluations. The inter-and intra-examiner Kappa coefficients for ICDAS ranged from 0.70 to 0.96. 

### 2.3. Ethical Aspects

This project was approved by the Research Ethics Committee (CEP) of the Federal University of Santa Maria (protocol CAAE 11765419.1.0000.5346). All guardians signed a written informed consent agreeing to have their son take part in the study. The adolescents also signed a consent form to participate in the study.

### 2.4. Data Analysis

Data were analysed using the STATA 14 (Stata Corporation, College Station, TX, USA). A descriptive analysis was carried out for the main sample characteristics. OHRQoL (CPQ11-14) were also described according to sociodemographic, social capital, and clinical variables. The characteristics of adolescents between those who completed the follow-up and dropouts from the original baseline sample were compared using the Chi-square test and the *t*-test for categorical and continuous variables, respectively. Similar tests were used to compare data of adolescents before and during the COVID-19 pandemic. The study outcome was the overall CPQ11-14 scores. All descriptive analysis was performed considering the sample weight by ‘svy’ command on STATA for complex data samples. The simple comparison between overall CPQ11-14 scores according to sample characteristics was performed using the Mann–Whitney test.

Multilevel Poisson regression analysis was performed to estimate the relationship of structural and cognitive social capital on the OHRQoL of adolescents. The construction of multilevel analysis considered adolescents (level 1) nested into 15 neighbourhoods (level 2). The analysis structure considered the fixed effect with the random intercept. Demographics (sex and age), socioeconomic (monthly household income), and the clinical variable (dental caries) were considered as possible confounders, as suggested by the previous literature on the predictors of OHRQoL [2,10]. Variables that presented *p* ≤ 0.20 in the unadjusted analysis were included in the adjusted model. The results are presented as Rate Ratio (RR) and 95% confidence intervals (95% CI).

## 3. Results

The sample comprised 429 adolescents, representing 67.1% of those evaluated at the cohort’s baseline. There were no statistical differences between the analysed sample and dropouts individuals for the evaluated characteristics (*p* > 0.05). Data obtained before and during the COVID-19 pandemic did also not differ (*p* > 0.05). 

Table 1 presents the descriptive characteristics of the adolescents. The sample was balanced between boys and girls, and the mean age was 12.6 years. Most individuals were from families with monthly household income > 1 BMW (70.6%). According to structural social capital variables, 45.2% and 95.1% of the adolescents attend religious meetings and visit friends and neighbours at least once a month, respectively. Considering cognitive social capital, 48.7% reported trusting in friends and neighbours, 57.8% believed that neighbours had good relationships, and 90.1% reported support during hard times.

OHRQoL (overall CPQ11-14) scores were reported according to sociodemographic, social capital, and clinical characteristics (Table 2). The mean of the overall CPQ11-14 scores was 11.2 (SE = 0.63). Higher OHRQoL scores were found in girls, individuals from families with low household income, and those who presented with dental caries. OHRQoL scores were also higher in individuals with lower structural and cognitive social capital considering all indicator variables. The highest OHRQoL scores were among participants reporting no support during hard times (mean =18.3). 

Table 3 presents the unadjusted and adjusted analysis between the social capital variables and OHRQoL scores. All independent variables were associated with OHRQoL (*p* < 0.05) in the unadjusted analysis. In the adjusted analysis, adolescents who attended religious meetings less than once a month or never (structural social capital) presented with OHRQoL scores 12% higher than their equivalents (RR 1.12; 95% CI 1.05–1.19). Considering the cognitive social capital variables, adolescents who did not trust their friends and neighbours (RR 1.10; 95% CI 1.03–1.17) and those who believed that their neighbours did not have good relationships (RR 1.06; 95% CI 1.01–1.14) were also more likely to present with worse OHRQoL. Adolescents reporting no one to support them during hard times presented with OHRQoL scores 24% higher than those who had this support (RR 1.24; 95% CI 1.11–1.37). In addition, female adolescents and those from low-income families and with dental caries were also more likely to report poor OHRQoL.

## 4. Discussion

This study aimed to evaluate the different roles of structural and cognitive dimensions of social capital on OHRQoL among adolescents. Our findings confirmed the hypothesis that both dimensions of social capital were associated with OHRQoL. Despite that the previous literature has evidenced the association between these social capital dimensions and oral health [11,12,13,14,15,16], the role of these factors in OHRQoL during adolescence has not been explored yet.

Overall, our findings showed that OHRQoL was poorer in individuals who presented with lower levels of structural and cognitive social capital. In this study, social capital variables were measured through indicators of social networks, social support, and trust, which have been considered proxy measures to assess social capital at the individual level [23,24]. In general, individuals reporting some source of social capital tend to have better general and oral health outcomes, as reported in previous studies [9,10,26,27,28]. This may be explained due to the fact that individuals with high social capital tend to experience positive ‘peer pressure’ to adopt healthy habits, increase the use of health services, and have less stress through the protective effects of social support [29], which may impact their oral health and quality of life.

Considering the structural dimension of social capital, our results showed that adolescents who attended religious meetings less than once a month or never presented with poorer OHRQoL than those who attended religious meeting more regularly. The structural social capital reflects the extent and intensity of participation in associations or other forms of social activity [3]. On the other hand, the frequency of visit to friends or neighbours were not related to OHRQoL in the adjusted analysis. This unexpected finding can be explained due to the fact that interpersonal relationships such as visiting friends and neighbours were very common among the participants. According to our results, almost the entire sample (95.1%) reported visits to friends and neighbours at least once a month, which may have influenced the results despite being considered an important proxy measure of social capital [23,24]. In contrast, attendance of religious meetings impacted adolescent’s OHRQoL. A previous study suggested that access to dental care and the adoption of healthy behaviour related to oral health are influenced by religiosity, which positively impacted individuals’ oral health [30]. It has also been shown that religiosity and the presence of churches in the neighbourhood are related to better oral health outcomes in children and adolescents [9,31,32]. Thus, this type of social tie may generate social resources that benefit the individual by improving social support and coping [29], which directly impacts subjective health outcomes, such as OHRQoL. 

According to our findings, all cognitive social capital indicators were related to OHRQoL scores. Adolescents who did not trust their friends and neighbours and who believed that their neighbours did not have good relationships with each other presented with worse OHRQoL. The cognitive social capital is considered the qualitative component of social capital and refers to people’s perception of interpersonal trust, solidarity, and reciprocity [3]. Previous studies have shown that high cognitive social capital was associated with better subjective health outcomes, such as mental health and self-perceived oral health [12,16], which is in accordance with our findings. These findings may be explained by the fact that high levels of interpersonal and neighbourhood trust may act as a protective factor that attenuates the negative effects of stress through feelings of security and belonging, [29,33] and, consequently, mitigates the impact of oral problems on an individuals’ quality of life. Our findings also demonstrated that adolescents who reported a lack of support during hard times presented with poorer OHRQoL. This cognitive indicator of social capital exerted the greatest impact on OHRQoL. Social support exerts a fundamental role during adolescence since it is related to other psychosocial factors, such as sense of coherence, self-management, and personal strength to deal with life problems [34]. These factors have a significant impact on oral health outcomes at this stage of life [35,36,37], where individuals experience biopsychosocial changes and emotional adaptations according to social structures [18,19,36]. Therefore, cognitive social capital seems to represent a relevant aspect of OHRQoL among adolescents.

Although this study focused on the different dimensions of social capital, it is important to emphasize the fact that there is a reciprocal relationship between the structural and cognitive dimensions of social capital. In this sense, studies have shown that social networks and social participation may lead to increased levels of social trust [6,38]. Likewise, it is reasonable to assume that individuals with high levels of social trust are more likely to socialize with others across different types of social networks [38]. Notwithstanding, despite the reciprocity among the social capital dimensions, our findings cautiously assume that, in adolescence, trust and social support (reflecting the cognitive social capital) operates as a central component in the composition of social capital.

This study has some limitations that need to be addressed. First, our study considered only some proxy variables of structural and cognitive dimensions of social capital. However, measuring all sources of social capital was out of the scope of this study. The social capital measures examined in this study have been used in previous research [7,9,10,23,24]. Furthermore, the losses to follow-up of adolescents between baseline and the studied sample may have impacted the external validity of our results. However, attrition analyses showed that the analysed participants were similar and representative to the baseline sample when the main characteristics were compared. Another limitation is due to some missing data in our database among followed individuals. However, this concern occurred in few cases and Bootstrap sensitivity analyses confirmed that missing data did not affect the validity of our results.

Despite the above-mentioned limitations, this study also has strengths that should be acknowledged, including the selection of a representative sample of the population of the city and the use of robust methods of data collection and analysis. Investigations of social capital and its different dimensions during adolescence is very relevant since the individuals build relationships and networks of trust during this period of life, which can impact their health throughout their life course. Moreover, this study also included patient-reported outcomes, which have been increasingly incorporated into epidemiologic studies in oral health. Public health policies to promote social capital in schools or communities are encouraged, since stimulating interpersonal social networks and social trust can directly benefit the general and oral health of adolescents and their living environment. Future longitudinal and interventional studies considering these factors are also encouraged to enhance the understanding of these interrelationships in more depth. 

## 5. Conclusions

Our findings suggest that OHRQoL was poorer in adolescents who presented with lower structural and cognitive social capital, with the greater impact on OHRQoL related to the cognitive dimension. These conclusions are important for public health policies to promote social capital, aiming to improve oral health and quality of life in this population.

## Figures and Tables

**Table 1 ijerph-20-05603-t001:** Demographic, socioeconomic, clinic, and social capital characteristics of the sample.

Variables	*n*	(%)
*Sociodemographic variables*		
Sex		
Boys	209	49.8
Girls	220	50.2
Age		
<12 years	189	43.4
≥12 years	240	53.6
Household income		
>1 BMW	264	70.6
<1 BMW	110	29.4
*Structural social capital*		
Attending religious meeting		
At least once a month	203	45.2
Less than once a month/never	226	54.8
Visit to friends/neighbours		
At least once a month	400	95.1
Less than once a month/ never	29	4.9
*Cognitive social capital*		
Trust in friends and neighbours		
Yes	202	48.7
No	223	51.3
Good relationships at neighbourhood		
Yes	240	57.8
No	189	42.2
Support during hard times		
Yes	381	90.1
No	38	9.9
*Oral health measures*		
Dental caries		
Absent	300	*69.4*
Present	128	*30.6*
*Outcome*	Mean	SE
CPQ11-14	11.2	0.6

Takes into account the sampling weight; values lower than 429 sample are due to missing data (trust in friends and neighbours (*n* = 4), support during hard times (*n* = 10), dental caries (*n* = 1), and household income (*n* = 55); SE, standard error; BMW, Brazilian minimum wages; CPQ, child perception questionnaire.

**Table 2 ijerph-20-05603-t002:** Overall OHRQoL (CPQ11-14) scores according to sociodemographic characteristics, social capital variables, and oral health measures.

Variables	CPQ11-14 Mean (SE)	*p*-Value *
*Sociodemographic variables*		
Sex		0.02
Boys	10.4 (0.8)	
Girls	11.9 (0.9)	
Age		0.08
<12 years	11.7 (0.7)	
≥12 years	10.7 (0.9)	
Household income		0.001
>1 BMW	11.2 (0.8)	
≤1 BMW	12.2 (0.9)	
*Structural social capital*		
Attending religious meeting		0.393
At least once a month	8.4 (0.9)	
Less than once a month/never	11.3 (0.6)	
Visit to friends/neighbours		0.459
At least once a month	10.1 (0.9)	
Less than once a month/ never	12.0 (0.8)	
*Cognitive social capital*		
Trust in friends and neighbours		0.05
Yes	10.3 (0.8)	
Not	12.0 (0.9)	
Good relationships at neighbourhood		0.02
Yes	10.9 (0.8)	
Not	11.5 (0.9)	
Support during hard times		0.175
Yes	10.3 (0.5)	
Not	18.8 (3.1)	
*Oral health measures*		
Dental caries		0.05
Absent	10.3 (0.6)	
Present	13.0 (1.3)	

Takes into account the sampling weight; SE, standard error; CPQ, child perception questionnaire; BMW, Brazilian minimum wages. * *p*-values refer to Mann–Whitney test.

**Table 3 ijerph-20-05603-t003:** Unadjusted and adjusted multilevel Poisson regression analysis between social capital dimensions and overall CPQ11-14 scores.

Variables	Unadjusted RR (95%CI)	*p*-Value	Adjusted RR (95%CI)	*p*-Value
*Sociodemographic variables*				
Sex		<0.001		<0.001
Boys	1 (reference)		1 (reference)	
Girls	1.23 (1.16–1.30)		1.28 (1.10–1.26)	
Age		0.037		0.341
<12 years	1 (reference)		1 (reference)	
>12 years	0.93 (0.88–0.99)		0.96 (0.90–1.03)	
Household income		<0.001		<0.001
>1 BMW	1 (reference)		1 (reference)	
≤1 BMW	1.27 (1.19–1.36)		1.17 (1.09–1.26)	
*Structural social capital*				
Attending religious meeting		0.019		<0.01
At least once a month	1 (reference)		1 (reference)	
	1.07 (1.01–1.13		1.12 (1.05–1.19)	
Visit to friends/neighbours		<0.01		0.113
At least once a month	1 (reference)		1 (reference)	
Less than once a month/ never	1.23 (1.08–1.39)		1.11 (0.97–1.27)	
*Cognitive social capital*				
Trust in friends and neighbours		<0.001		<0.01
Yes	1 (reference)		1 (reference)	
No	1.14 (1.07–1.21)		1.10 (1.03–1.17)	
Good relationships at neighbourhood		<0.001		<0.05
Yes	1 (reference)		1 (reference)	
No	1.15 (1.08–1.22)		1.06 (1.01–1.14)	
Support during hard times		<0.001		<0.001
Yes	1 (reference)		1 (reference)	
No	1.21 (1.10–1.34)		1.24 (1.11–1.37)	
*Oral health measures*				
Dental caries		<0.001		<0.001
Absent	1 (reference)		1 (reference)	
Present	1.16 (1.09–1.24)		1.14 (1.06–1.23)	

RR, rate ratio; CI, confidence interval; CPQ, child perception questionnaire; BMW, Brazilian minimum wages.

## Data Availability

Data available on request from the authors.

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
