# Peer review of "The Different Roles of Structural and Cognitive Social Capital on Oral Health-Related Quality of Life among Adolescents"

_ijerph, 2023, doi:10.3390/ijerph20085603_

Round 1
Reviewer 1 Report
I was pleased to review the manuscript “ijerph-2270015” entitled “The different roles of structural and cognitive social capital on oral health-related quality of life among adolescents” for the International Journal of Environmental Research and Public Health. This study assessed the relationship of structural and cognitive social capital with oral health-related quality of life among adolescents, which was steel needed considering that the literature provides more information about this topic on adult and elderly population, but not as much for adolescents. Overall, it is a well-structured study with adequate and sound methods, that provides a good background about oral health-related quality of life and structural and cognitive capital. It also discusses very well the results based on previous and appropriate studies and highlights the importance of the outcomes in the life of the adolescents. Furthermore, the conclusions are consistent with the findings presented and discussed, and answer the main question proposed by the authors. This study is relevant to the field since its outcomes can help to understand and develop new strategies to promote oral health and quality of life in adolescents and hopefully succeed in carrying these improvements for the rest of their lives, considering that adolescence is a period of our life in which social relationships and personal growth are established, having an important role in the decisions and way of life that people will choose when become adults. However, I suggest minor revision to facilitate and improve the reading.
Please refer to comments below:
Check all manuscript for typos. Example: the second paragraph in the Methods section says “90% of children of in the city.”
Abstract:
Please remove the headings.
The abstract seems long. Please try to summarize it.
Discussion:
I suggest including a brief comment about how the results can affect/impact directly the life of these adolescents? What kind of public health policies could be implemented to help these adolescents? What effects it can bring to society?
Author Response
Reviewer: 1
I was pleased to review the manuscript “ijerph-2270015” entitled “The different roles of structural and cognitive social capital on oral health-related quality of life among adolescents” for the International Journal of Environmental Research and Public Health. This study assessed the relationship of structural and cognitive social capital with oral health-related quality of life among adolescents, which was steel needed considering that the literature provides more information about this topic on adult and elderly population, but not as much for adolescents. Overall, it is a well-structured study with adequate and sound methods, that provides a good background about oral health-related quality of life and structural and cognitive capital. It also discusses very well the results based on previous and appropriate studies and highlights the importance of the outcomes in the life of the adolescents. Furthermore, the conclusions are consistent with the findings presented and discussed, and answer the main question proposed by the authors. This study is relevant to the field since its outcomes can help to understand and develop new strategies to promote oral health and quality of life in adolescents and hopefully succeed in carrying these improvements for the rest of their lives, considering that adolescence is a period of our life in which social relationships and personal growth are established, having an important role in the decisions and way of life that people will choose when become adults. However, I suggest minor revision to facilitate and improve the reading.
Please refer to comments below:
- Check all manuscript for typos. Example: the second paragraph in the Methods section says “90% of children of in the city.”
Response: Thank you for your comment. The manuscript has been checked for solving possible typos and slips.
Abstract:
- Please remove the headings.
Response: Thank you for your comment. The headings have removed.
- The abstract seems long. Please try to summarize it.
Response: Thank you for your comment. The abstract has been adjusted as suggested: “To evaluate the relationship of structural and cognitive dimensions of social capital with oral health-related quality of life (OHRQoL) among adolescents. This was a cross-sectional study nested in a cohort of adolescents from southern Brazil. OHRQoL was evaluated using the short version of Child Perceptions Questionnaire 11-14 (CPQ11-14). Structural social capital was measured by attendance of religious meetings and social networks from friends and neighbours. Cognitive social capital was evaluated through trust in friends and neighbours, perception of relationships in the neighbourhood and social support during hard times. Multilevel Poisson regression analysis was performed to estimate the association between social capital dimensions and overall CPQ11-14 scores - higher scores correspond to worse OHRQoL. The sample com-prised 429 adolescents with a mean age of 12 years. Adolescents who attended religious meetings less than once a month or never presented higher overall CPQ11-14 scores. Adolescents who did not trust their friends and neighbours, those who believe that their neighbours did not have good relationships, and those reporting no support during hard times also presented higher overall CPQ11-14 scores. OHRQoL was poorer in individuals who presented lower structural and cognitive social capital, with the greatest impact related to the cognitive dimension.”
Discussion:
- I suggest including a brief comment about how the results can affect/impact directly the life of these adolescents? What kind of public health policies could be implemented to help these adolescents? What effects it can bring to society?
Response: Thank you for your comment. This brief comment has been added in the discussion as suggested: “Public health policies to promote social capital in schools or communities are encouraged, since stimulating interpersonal social networks and social trust can directly benefit the general and oral health of adolescents and their living environment. Future longitudinal and interventional studies considering these factors are also encouraged to enhance the understanding in more depth these interrelationships.”

Reviewer 2 Report
Dear authors!
The presented article is aimed at evaluation of roles of structural and cognitive social capital on oral health-related quality of life among adolescents. The selection of adolescents as an experimental group is very interesting because this age period is connected with biopsychosocial changes, shaping the worldview and interpersonal relationships. It is worth noting that all the processes mentioned above, associated with adolescents, can affect the condition of a person's oral cavity and the quality of his life. Thus, the issue raised in your study is really very interesting.
The text of the study is well structured, the content is clearly stated. The information presented in the "Methods" section is described in sufficient detail to repeat the experiment. The results of the study are presented graphically in the form of tables.
The presented work leaves a positive impression, however, there is some suggestions and questions:
1. There is a certain file format in the requirements to manuscript on the journal’s website. It is worth paying attention to the template posted on the publisher’s website containing a line numbering which greatly facilitates understanding and communication between authors and reviewers. https://www.mdpi.com/journal/jpm/instructions#:~:text=Accepted%20File%20Formats,for%20further%20details).
2. In the abstract, the aim of the study is specified in the "Background" subsection. Please add a brief information about the relevance of the research topic on the basis of which the purpose of the study was formulated.
3. In the abstract, in the “Results” subsection, a closing parenthesis in the part of the text « (standard error [SE] = 0.1] » should be corrected.
4. In the study of 2010 (literary reference â„–20) is noticed that prior to the examination of dental caries, extensive examiner training was performed and in total seventeen examiners (graduate dental students) participated in the study. Please, specify, were the examiners in the presented study of 2020 the same? If not, add information about the criteria of examiners selection in the “Method” section.
5. In the Table 1 you indicated that values lower than 429 sample are due to missing data. Please, provide more information about this in the “Discussion” section.
Author Response
Reviewer: 2
Dear authors! The presented article is aimed at evaluation of roles of structural and cognitive social capital on oral health-related quality of life among adolescents. The selection of adolescents as an experimental group is very interesting because this age period is connected with biopsychosocial changes, shaping the worldview and interpersonal relationships. It is worth noting that all the processes mentioned above, associated with adolescents, can affect the condition of a person's oral cavity and the quality of his life. Thus, the issue raised in your study is really very interesting.The text of the study is well structured, the content is clearly stated. The information presented in the "Methods" section is described in sufficient detail to repeat the experiment. The results of the study are presented graphically in the form of tables.
The presented work leaves a positive impression, however, there is some suggestions and questions:
- There is a certain file format in the requirements to manuscript on the journal’s website. It is worth paying attention to the template posted on the publisher’s website containing a line numbering which greatly facilitates understanding and communication between authors and reviewers. https://www.mdpi.com/journal/jpm/instructions#:~:text=Accepted%20File%20Formats,for%20further%20details).
Response: Thank you for your comment. The journal's guidelines have been checked. The current template used in the manuscript has been provided by the journal.
- In the abstract, the aim of the study is specified in the "Background" subsection. Please add a brief information about the relevance of the research topic on the basis of which the purpose of the study was formulated.
Response: Thank you for your comment. The reviewer 1 requested to remove the headings and also to shorten the abstract (up to 200, according to journal’s guidelines). For that reason, the subheadings have been removed, and extra information has not been added: “To evaluate the relationship of structural and cognitive dimensions of social capital with oral health-related quality of life (OHRQoL) among adolescents. This was a cross-sectional study nested in a cohort of adolescents from southern Brazil. OHRQoL was evaluated using the short version of Child Perceptions Questionnaire 11-14 (CPQ11-14). Structural social capital was measured by attendance of religious meetings and social networks from friends and neighbours. Cognitive social capital was evaluated through trust in friends and neighbours, perception of relationships in the neighbourhood and social support during hard times. Multilevel Poisson regression analysis was performed to estimate the association between social capital dimensions and overall CPQ11-14 scores - higher scores correspond to worse OHRQoL. The sample com-prised 429 adolescents with a mean age of 12 years. Adolescents who attended religious meetings less than once a month or never presented higher overall CPQ11-14 scores. Adolescents who did not trust their friends and neighbours, those who believe that their neighbours did not have good relationships, and those reporting no support during hard times also presented higher overall CPQ11-14 scores. OHRQoL was poorer in individuals who presented lower structural and cognitive social capital, with the greatest impact related to the cognitive dimension.”
- In the abstract, in the “Results” subsection, a closing parenthesis in the part of the text « (standard error [SE] = 0.1] » should be corrected.
Response: Thank you for your observation. This part of abstract has been removed to reduce the number of words.
- In the study of 2010 (literary reference â„–20) is noticed that prior to the examination of dental caries, extensive examiner training was performed and in total seventeen examiners (graduate dental students) participated in the study. Please, specify, were the examiners in the presented study of 2020 the same? If not, add information about the criteria of examiners selection in the “Method” section.
Response: Thank you for your comment. The baseline examiners were not the same as the follow-up examiners. In both cohort waves, the clinical examiners comprised dentists which were previously trained and calibrated for ICDAS evaluation. A total of 17 and 7 examiners conducted clinical examinations at baseline and follow-up, respectively. In the present study, only follow-up dental caries assessment data were used. The reference "20", at the beginning of the methods section was used just to contextualize the original sample of this cross-sectional study nested within cohort. The training and calibration process of 2020 evaluation has been clarified in the methods section: “A total of seven examiners conducted clinical examinations. All clinical examiners were dentists which were previously trained and calibrated for ICDAS evaluation, totalling a process of 36 hours, including theoretical training, photographic image evaluation, exercise with exfoliated teeth and clinical evaluations. The inter-and intra-examiner Kappa coefficients for ICDAS ranged from 0.70 to 0.96.”
- In the Table 1 you indicated that values lower than 429 sample are due to missing data. Please, provide more information about this in the “Discussion” section.
Response: Thank you for your comment. Indeed, missing data was observed for few participants in the following variables: trust in friends and neighbours (n=4), support during hard times (n=10) and dental caries (n=1). Missing data for household income was higher (n=55). This fact is common since individuals sometimes feel embarrassed to report their income. This information has been inserted in the footnote of #Table “Values lower than 429 sample are due to missing data [trust in friends and neighbours (n=4), support during hard times (n=10), dental caries (n=1) and household income (n=55)].”
Nonetheless, Bootstrap sensitivity tests were performed to confirm that these missing data did not influence our results, being considered acceptable from the statistical point of view. This concern has also been discussed in the study limitations: “Another limitation is due to some missing data in our database among followed individuals. However, this concern occurred in few cases and Bootstrap sensitivity analyses confirmed that missing data did not affect the validity of our results.”

Reviewer 3 Report
I was pleased to read the manuscript entitled "The different roles of structural and cognitive social capital on oral health-related quality of life among adolescents" and to review it.
The study examines the relationship of structural and cognitive dimensions of social capital with oral health-related quality of life (OHRQoL) among adolescents. The article by its content is really suitable for the special issue "Multidimensional Aspects of Oral Health-Related Quality of Life".
The article is written in a typical format, it is structured logically and the text is fluent. The rationale of the study is well described and the study problem is stated clearly. The reliability of the collected data and analysis methods is beyond doubt. The conclusions are supported by the presented results. However, I would like to suggest to the authors to draw attention to some points.
Abstract – It would be good to indicate that higher CPQ scores correspond to worse OHRQoL. Remove titles (Background, Methods, ...).
Methods –Cite a post hoc power calculation instrument. The sentence "The overall CPQ11-14 scores were used in the data analysis and can range from 0 to 64 points, with higher scores indicating a greater impact on quality of life." should be corrected as follows: "The overall CPQ11-14 scores were used in the data analysis and can range from 0 to 64 points, with higher scores indicating a lower level of quality of life."
Results – Table 2 and corresponding text: indicate p-values for differences in means.
Table 3: give p-values for adjusted RR in the same way as for unadjusted RR.
Thank you for considering my opinion. I encourage authors to keep on working to improve the manuscript.
Author Response
Reviewer: 3
I was pleased to read the manuscript entitled "The different roles of structural and cognitive social capital on oral health-related quality of life among adolescents" and to review it. The study examines the relationship of structural and cognitive dimensions of social capital with oral health-related quality of life (OHRQoL) among adolescents. The article by its content is really suitable for the special issue "Multidimensional Aspects of Oral Health-Related Quality of Life".
The article is written in a typical format, it is structured logically and the text is fluent. The rationale of the study is well described and the study problem is stated clearly. The reliability of the collected data and analysis methods is beyond doubt. The conclusions are supported by the presented results. However, I would like to suggest to the authors to draw attention to some points.
Abstract – It would be good to indicate that higher CPQ scores correspond to worse OHRQoL. Remove titles (Background, Methods, ...).
Response: Thank you for your comment. This information has been included in the abstract: “Multilevel Poisson regression analysis was performed to estimate the association between social capital dimensions and overall CPQ11-14 scores - higher scores correspond to worse OHRQoL.”
Methods –Cite a post hoc power calculation instrument.
Response: The instrument used has added, as follows: “The power of the study was calculated in OpenEpi [21] through a post hoc power calculation comparing adolescents' CPQ11-14 scores between the non-exposed group (high social capital) and exposed group (low social capital) according to social capital variables. Considering an alpha error probability of 0.05 and a 95% confidence interval, the study power ranged from 80 to 100%.”
[21] Dean AG, Sullivan KM, Soe MM. OpenEpi: Open Source Epidemiologic Statistics for Public Health, Version. www.OpenEpi.com, 2013/04/06, accessed 2023/04/10.
The sentence "The overall CPQ11-14 scores were used in the data analysis and can range from 0 to 64 points, with higher scores indicating a greater impact on quality of life." should be corrected as follows: "The overall CPQ11-14 scores were used in the data analysis and can range from 0 to 64 points, with higher scores indicating a lower level of quality of life."
Response: The sentence has been replaced as suggested.
Results – Table 2 and corresponding text: indicate p-values for differences in means.
Response: The p-values have been added as suggested. Please see #Table 2.
Table 3: give p-values for adjusted RR in the same way as for unadjusted RR.
Response: The p-values have been added as suggested. Please see #Table 3.
Thank you for considering my opinion. I encourage authors to keep on working to improve the manuscript.
Response: Thank you for all suggestions aiming to improve our manuscript.
